# Sonographic Anatomy and Normal Measurements of the Human Kidneys: A Comprehensive Review

**DOI:** 10.3390/diagnostics15243208

**Published:** 2025-12-15

**Authors:** Madhvi Yadav, Saubhagya Srivastava, Manjiri Dighe, Kathleen Möller, Christian Jenssen, Christoph Frank Dietrich

**Affiliations:** 1Department of Radiology, University of Washington, Seattle, WA 98195, USA; madhvi@uw.edu (M.Y.); ssaubhagya.s@gmail.com (S.S.); dighe@uw.edu (M.D.); 2Medical Department I/Gastroenterology, SANA Hospital Lichtenberg, 10365 Berlin, Germany; k.moeller@live.de; 3Department for Internal Medicine, Krankenhaus Märkisch Oderland, 15344 Strausberg, Germany; c.jenssen@khmol.de; 4Brandenburg Institute for Clinical Ultrasound (BICUS), Brandenburg Medical University, 16816 Neuruppin, Germany; 5Department General Internal Medicine (DAIM), Hospitals Hirslanden Bern Beau Site, Salem and Permanence, Schänzlihalde 11, 3013 Bern, Switzerland

**Keywords:** kidney/ultrasonography, kidney diseases/diagnostic imaging, renal artery/ultrasonography, echogenicity, renal Doppler

## Abstract

Ultrasound is the primary, non-invasive imaging modality for evaluating renal anatomy and function in both acute and chronic settings. Familiarity with normal kidney morphology, cortical and parenchymal thickness, echogenicity, and Doppler parameters is essential for differentiating normal findings from early manifestations of disease. This review summarizes established reference ranges and anatomical variants from the 1950s to 2025, highlighting differences related to age, sex, body habitus, and ethnicity. Practical emphasis is placed on the interpretation of renal size, cortical thickness, echogenicity, and resistive indices in clinical scenarios such as chronic kidney disease, renovascular hypertension, acute obstruction, and renal transplantation. By integrating sonographic measurements with clinical and laboratory findings, clinicians can achieve timely diagnosis, monitor disease progression, and guide therapeutic decisions while minimizing the need for invasive or radiation-based imaging.

## 1. Introduction

Ultrasound (US) plays a crucial role in the initial evaluation of suspected renal pathologies when patients initially present with symptoms such as hematuria, flank pain, or an increasing creatinine level. It can reveal renal morphology, physical features, and potential abnormalities. Ultrasound holds several advantages, including but not limited to being cost-efficient, non-invasive, and devoid of ionizing radiation. The current advancements in miniaturization, portability, and affordability of the equipment have facilitated the point-of-care sonographic evaluation by clinicians as well [1]. Ultrasound is also utilized in guiding biopsies and vascular access for temporary or permanent hemodialysis in patients with renal dysfunction [2].

While conventional ultrasound is often adequate to classify indeterminate renal lesions as benign in simple or minimally complex cystic masses, conventional US may not be reliable enough in defining cystic masses with more complex features such as septae or nodularity. More novel and advanced techniques like contrast-enhanced US (CEUS) have been able to address some of the limitations of conventional B-mode and Doppler US [3]. Ultrasound elastography is an emerging technique that allows non-invasive assessment of renal tissue stiffness. The existing literature reports varied and heterogeneous stiffness cut-offs and acquisition protocols [4]. Although elastography holds a promising future, it warrants further investigation. A detailed discussion is beyond the scope of this review paper.

Knowledge of quantitative measures such as the renal cortical thickness and subjective measures such as the echogenicity, are crucial for differentiating normal from pathological findings. These measurements vary based on age, sex, body habitus, and establishing population-specific reference ranges can broaden the understanding of normal anatomy and potential variations. US findings can occasionally be nonspecific hence, it is important to correlate them clinically and with laboratory parameters, including serum creatinine, estimated glomerular filtration rate (eGFR), blood urea nitrogen (BUN) and urinalysis.

## 2. Methods of Literature Review

A narrative review was performed with systematic literature search using multiple electronic databases such as PubMed, Scopus, Web of Science, EMBASE, and others using key words such as “normal kidney ultrasound findings”, “normal kidney ultrasound”, “renal doppler”, “renal medulla”, “renal resistive index”, “renal mass”, “renal echogenicity”, and “urolithiasis” (kindly review the Appendix A for detailed Boolean structure). The search was conducted from March 2024–May 2025 for published literature from the 1950s and onwards. The literature search was conducted in the English language. Relevant articles were further hand-searched for additional relevant literature. All the literature that was based on non-human or veterinarian subjects, unpublished or opinion articles, and conference abstracts that lacked methodological details, was excluded. Articles reporting normal anatomical ranges, physiological parameters, and pathological findings were included in the study. In two sequential steps, study selection was carried out, firstly reviewing articles by title and abstract, and secondly by assessing the full text of the articles. The literature was expanded by further reviewing references from included articles. Figure 1 outlines the process for study selection

In our literature review we extracted reference normal ranges, when available, along with key details such as: (a) Demography of the study population, such as age distribution, sex, body habitus or BMI category, and other relevant characteristics; (b) Technical imaging parameters, such as ultrasound system model and software version, type of transducer, and frequency, patient positioning, and measurement plane; and (c) Statistical measures and uncertainty metrics, including 95% confidence intervals, and intra- or interobserver variability. This information guided our narrative synthesis and, where possible, summarized in the table. In this review, we did not generate or analyze any new patient-level information. The search strategy and study selection process are summarized in the flow diagram.

## 3. Normal Gross Anatomy

The right kidney is placed adjacent to the liver and the left kidney is placed adjacent to the spleen; these organs serve as an acoustic window for better visualization of the kidneys. The kidneys lie in the retroperitoneum and have three key compartments—(i) perirenal space (enclosed by Gerota’s fascia; contains the kidneys, adrenal glands, and perirenal fat), (ii) anterior pararenal space (lies between the anterior pararenal fascia and the posterior peritoneum), and (iii) posterior pararenal space (lies between the posterior pararenal fascia and the transversalis fascia) (Figure 2).

On the US, these fasciae cannot be visualized, and thus, the peri- and para-renal spaces appear as single fat-filled compartments. When water content is reduced, perirenal fat may appear hypoechoic and may mimic perirenal fluid collections [5].

On the right side, the space between the kidney and the liver is normally 1–6 mm thick, and it is ≤1 mm in approximately half of the patients. Obesity can lead to an increase in this distance due to the accumulation of perirenal fat while aberrant veins may occasionally be mistaken for fluid collections.

Changes in the echogenicity and size of the anterior pararenal space can also be indicative of various pathological conditions. Acute abdominal inflammation (due to conditions such as acute pancreatitis, acute appendicitis, ischemic bowel, liver abscess, or penetrated duodenal ulcer) can cause increased echogenicity and widening of this space [6]. In contrast, conditions like carcinoma, lymphoma, and chronic inflammation may enlarge the anterior pararenal space without altering its echogenicity.

## 4. Intrarenal Anatomy

The internal structure of a kidney is composed of multiple lobules, each consisting of an outer rim of cortical tissue surrounding a medullary pyramid that terminates at papillae, which then extends into minor calyx. Calyx is the cavity of the kidney through which urine passes and consists of minor and major calyces. The minor calyces converge into the major calyces, which in turn form the renal pelvis and continue as the proximal ureter. The urinary tract is lined with urothelium, which is less than 2 mm thick. In cases of acute pyelonephritis, the urothelium may thicken to >2 mm [7]. The area between the calyces and renal parenchyma is known as the renal sinus. This space contains blood vessels, lymphatics, and fat tissue, which contributes to its hyperechoic appearance on the US. The cortex appears hypoechoic or isoechoic relative to the liver (on the right side) and the spleen (on the left side). Whereas the pyramids appear slightly less echogenic than the cortex (Figure 3). The extension of the cortex into the medulla is called the Column of Bertin. The renal columns are named after the French physician and anatomist Exupère Joseph Bertin (1712–1781) [8]. Sometimes, it can be confused with a neoplasm if they are hypertrophied. This can be ruled out by looking for the continuity of the renal column of Bertin with the cortex, the preservation of the cortical outline, and normal vasculature in this region, which is not disrupted in its architecture. Typically, the renal columns are bordered on both sides by a medullary pyramid. The renal hilum, which includes the vascular pedicle supplying the kidney, is positioned adjacent to the renal pelvis but lies outside the sinus [5,9].

### 4.1. Renal Vascular Anatomy

The kidneys are generally supplied by one renal artery arising from the aorta, one vein draining into the IVC, and receive about 20% of the cardiac output [10]. Radiological evaluation of renal arteries is central to diagnosis of conditions such as renal artery stenosis, fibromuscular dysplasia, and vascular malformations. Understanding various variants such as accessory renal arteries, aberrant branches, or early bifurcation impacts the technical complexity of renal transplantation, influences graft perfusion, and hence the success of the transplant.

#### 4.1.1. Renal Artery

Renal artery originates from the abdominal aorta just below the superior mesenteric artery at the level of L1–L2. Anatomical variations are common, with accessory renal artery present in up to 30% of individuals unilaterally, and in 10% it may be present bilaterally. The main renal artery is 4–6 cm long and 5–6 mm in diameter [11]. The right renal artery is located higher than the left renal artery. It is also longer than the left and courses posteriorly to the inferior vena cava (IVC). The left renal artery courses almost horizontally to the left kidney and posteriorly to the renal vein. In approximately 30% of the cases, renal arteries can be located anterior to the renal veins [12]. Smaller branches are given out to the surrounding organs during the course. The caliber of these branches is small and generally not easily appreciated on the US. The main renal artery divides into five segmental branches before entering the renal hilum. The segmental branches include—apical, superior, middle, inferior, and posterior segmental arteries. The segmental arteries are end arteries, and vascular insult in these segmental arteries could lead to permanent renal damage in their respective territories. They further divide into the lobar, interlobar, arcuate, and interlobular arteries.

*Anatomical variants*: The majority of existing literature describes the accessory renal artery as an artery arising from the abdominal aorta in addition to the main renal artery [13,14,15]. It can either pass through the hilum or directly terminate at the poles to supply the kidneys. It is seen in approximately 25–30% of the patients [11]. It is noted that the accessory artery to the upper pole is typically smaller in diameter than those at the lower poles. Graves described an aberrant artery in a patient where it reached the parenchyma of the kidney but originated from a source other than the aorta [16]. Pollak et al. detected about 23% of double renal arteries, 4% triple renal arteries, and 1% quadruple renal arteries in a 400 cadaveric renal donors with 800 kidneys [17]. There are reported cases of bilateral multiple renal arteries seen in approximately ~15% of the population [18].

A variant may be classified as a prehilar artery (early branching of the renal artery) when a branch diverges within 1.5–2.0 cm of origin in the left renal artery or in the retrocaval segment of the right renal artery. It has been seen that prehilar branching shows high individual variability. Ozkan et al. reported prevalence of 8% of prehilar renal arteries in his study [19]. According to the number of pre-segmental arteries, the branching pattern of a single main renal artery can be classified as Type I (segmental arteries without pre-segmental arteries), Type II (segmental arteries plus one pre-segmental artery), and Type III (segmental arteries plus two pre-segmental arteries). Further, they are sub grouped according to the number of segmental arteries [20]. This knowledge is significant, especially during renal transplantation, as surgeons require at least 1.5 to 2.0 cm of renal artery before first branching, for successful anastomosis [11].

#### 4.1.2. Renal Vein

The renal vein generally lies anterior to the renal artery in the renal hilum. The left renal vein (6–10 cm) is significantly longer than the right renal vein (2–4 cm) [21]. Before entering the IVC medially, the left renal vein passes between the aorta and the superior mesenteric artery. Left renal vein drains blood from the left adrenal and gonadal veins and, in most patients, from the lumbar veins as well. The shorter right renal vein empties into the IVC laterally. It is noted that in comparison to the left renal vein, only 7% of the right gonadal and 31% of the adrenal veins drain into the right renal vein [11,12,22].

*Anatomical variants*: As compared to the arteries, variation in renal venous anatomy is less common. The most common variant is multiple renal veins, seen in 15–30% of the population [23,24]. It is more common on the right side than the left. Late venous confluence is another variant. On the left side it is diagnosed when the venous branches join within 1.5 cm of the left lateral wall of abdominal aorta, or when the branches join within 1.5 cm of their confluence with the IVC. Circumaortic left renal vein (consisting of anterior and posterior limbs that encircle the abdominal aorta) is another common congenital anomaly seen in 2–17% of the population [8].

## 5. Ultrasound Evaluation of Renal Parenchyma and Vasculature

### 5.1. B-Mode Ultrasound (Grayscale Imaging)

#### Imaging Techniques and Protocol

*Patient positioning:* The examination typically begins with the patient in the supine position. To enhance comfort and facilitate optimal imaging, especially in patients with a low-lying kidney or increased body habitus, a pad, pillow, or rolled towel can be placed under the flank or beneath the upper shoulders. This support aids in elevating the area of interest and may improve acoustic windows. In cases where the supine position does not provide adequate visualization due to factors like overlying bowel gas, repositioning the patient to a lateral decubitus or oblique position can be beneficial [5].

*Transducer selection:* Low-frequency curved array transducers with a frequency range of 1–6 MHz are used in adult patients. In situations where kidneys are present more superficially (e.g., pediatric kidneys or transplanted kidneys), linear array higher frequency transducers (>9 MHz) are used instead of lower frequency curved array transducers [10,25].

*Imaging technique:* The scanning technique involves acquiring images in both longitudinal and transverse planes of the kidney. For the longitudinal view, the probe is positioned in the mid-axillary line at the tenth intercostal space. After the long axis is viewed, the probe should be fanned anteriorly and posteriorly to image it completely. The probe is then rotated 90 degrees to obtain the short axis of the transverse view. The right kidney is best visualized in the anterior or midaxillary line, with the liver serving as an acoustic window. Visualization of the left kidney is more challenging since it lies beneath the spleen and potentially interface with the bowel gas. Adequate visualization requires imaging from the midaxillary or posterior axillary line. Real-time cine clips can be valuable for documenting and assessing renal perfusion or detecting calculi. It is recommended to examine the kidneys in several positions, e.g., in supine and lateral decubitus positions and sometimes also in prone position. This allows the entire kidney to be viewed and ensures that no findings are overlooked. Additionally, evaluation of adjacent structures, such as the bladder and proximal ureters, to assess for hydronephrosis or other related pathologies can be performed [4] (Table 1, Table 2 and Table 3, Figure 4, Figure 5 and Figure 6).

## 6. Shape and Appearance

On the US, a normal kidney exhibits an oval shape in the sagittal plane. The sinus fat is visualized as a bright echogenic center, bordered by a hypoechoic rim, representing the cortex and medulla. The renal sinus contains the pyelon, vessels, connective tissue, and fatty tissue. In the transversal transducer position, there is a noticeable interruption in the parenchymal rim at the level of the renal pelvis and hilum. Due to the kidney’s transversal width being approximately 27% greater than its sagittal diameter, it appears fuller or more “plump” on transversal images. This anatomical variance explains why the right kidney, typically imaged in a sagittal transducer position, appears thinner compared to the left kidney. The calyceal system is usually not visible unless distended with urine, which can enhance its visualization on the US. Similarly, it is difficult to visualize intrarenal vessels due to their smaller caliber and the presence of sinus fat, which masks their visibility. Occasionally, arcuate vessels may be visualized as hyperechoic dots at the junction of the cortex and medulla, along the outer edges of the medullary pyramids. For accurate longitudinal imaging, the kidney should present a uniform rim of parenchyma surrounding the echogenic sinus fat. The cortex may appear slightly thicker at the poles, and any irregularities in this rim could indicate an oblique imaging angle rather than true structural abnormalities. The appearance of neonatal kidneys differs significantly from that of older children and adults. Neonatal kidneys are smaller, with a more echogenic and lobulated cortex, while the medullary pyramids are more prominent and readily distinguishable. These differences are important to recognize, as they represent normal developmental anatomy rather than pathological findings [4]. The lobulation of the surface in this context is called renculation. During fetal development, the kidney forms through the fusion of several fetal reniculi. The reniculi consist of a medulla and a cortex. As a result of the fusion of the reniculi, a superficial contour indentation may remain evenly distributed between the reniculi in newborns and infants. In adults, renculation of the surface is very rare and should not be misinterpreted as parenchymal scarring.

## 7. Size

The size of the kidney is measured from pole to pole (bipolar length). This bipolar length is the best measure of kidney size because it can be easily obtained and is reproducible [26]. It measures between 10–12 cm [9].

In 2012, Sienz et al. [27] analyzed data on kidney size from various studies in a meta-analysis. They found that in most studies, the left kidney was larger than the right kidney. While kidney size did not differ significantly in healthy individuals aged 30–60, kidney size and volume decreased from the age of 60 onwards and became smaller with each decade of life. Kidney length and volume also correlated with height, surface area, and body mass index. Gender differences were described in most studies. Males had larger kidney size parameters and kidney volume than females. However, it remained unclear whether this was due to anthropomorphic parameters. While blood pressure had no effect on kidney size, patients with diabetes mellitus had larger kidneys. The measurement of longitudinal kidney size had lower inter- and intraobserver variability than kidney width and kidney volume. This meta-analysis, which included measurements from Malaysia, South India, Pakistan, Jamaica, Nigeria, South Korea, the US, Belgium, Germany, Great Britain, Croatia, Finland, Italy, and Turkey, also showed that ethnic data with different body types must be taken into account [27].

Harmse et al. demonstrated a relationship between body habitus and renal size using the following prediction model: Kidney length (mm) = 49.18 + 0.21 × weight (kg) + 0.27 × height (cm). Although this model has been tested with CT scans, reproducibility with US has not been evaluated. Based on CT measurements, it was found that the renal length best correlates with the body height in centimeters (cm) [28].

The absolute renal lengths of both kidneys in males are significantly larger than in females, which could be due to anthropometric variations [29]. Milétic et al. proposed that relative renal length (kidney length: kidney–body height ratio) could be used as an alternative parameter to renal length due to its lesser variability with body habitus, age, and sex compared to renal length [30]. In the pediatric population, height was identified as a significant predictor of kidney length [31].

Emmamian et al. and Miletic et al. showed a significant decrease in absolute and relative renal length in the age group 60–69 years old. They also showed that renal length gradually decreases with age, and this decrease accelerates after the age of 60 years [25,26]. Physiologically, renal length decreases 0.5 cm per decade after middle age [32]. Conversely, certain pathologies can lead to an increase in kidney size, including early-stage renal vein thrombosis, early-stage diabetes mellitus, and renal inflammation. A physiological increase in glomerular filtration rate can lead to an increase in kidney size, as observed in pregnancy [33,34]. Subjects with congenital single kidney or after unilateral nephrectomy have larger kidney lengths and volumes of the remaining kidney [35].

After living donor transplantation, kidney size and volume of both the remaining and transplanted kidneys increased [36,37].

The right kidney is smaller than the left kidney, with median lengths being 10.9 ± 1.2 and 11.2 ± 1.3 cm, respectively [26]. The discrepancy in length between the normal right and left kidney is not more than 1.5–2.0 cm [33]. Several hypotheses have been proposed to explain this discrepancy, including the presence of the liver on the right side with less spatial growth of the corresponding right kidney and a greater amount of blood flow to the left kidney due to an anatomically shorter left renal artery [38]. Additionally, body habitus and build are major predictors of renal size in healthy adults and children. In a study by Khan et al. [39] involving 2212 subjects, there was a significant negative correlation between age and kidney size. The maximum kidney length was measured in the 31 to 40 age group, with a decline in size from the age of 60 onwards also observed in this study. The authors attributed this to an age-related decline in kidney function in patients over 60 years of age. In our view, older patients have comorbidities such as diabetes mellitus, arteriosclerosis, and arterial hypertension. However, participants with arterial hypertension and diabetes mellitus were excluded from the study. In this study, too, the kidneys were significantly larger in men than in women, and there was a significant difference between different BMI categories. There was a significant difference (each *p* < 0.001) in the size of the right and left kidneys between underweight and overweight individuals, underweight and obese individuals, individuals with normal BMI and obese individuals, and overweight and obese individuals [39]. The study by Tiryaki et al. [40] involving 1918 subjects also confirmed that men’s kidneys are larger than women’s and that the left kidney is larger than the right. However, no correlation with BMI was found [40]. Although there are differences in size between the genders and between the right and left kidneys, these differences are within the normal reference values.

*Ethnic variation in kidney size*: Given the current burden of kidney diseases, with one in ten adults suffering from kidney problems, it is essential to have standardized kidney dimensions for assessing renal function, monitoring disease progression, guiding prognostic evaluations, and future follow-ups. The current renal length/size reference parameters are derived from the Caucasian or Western population. Multiple studies have demonstrated measurable differences in renal dimensions, particularly within Asian populations. Applying a generalized standard for renal size can lead to a misdiagnosis of the pathology, resulting in unnecessary follow-up imaging and increased patient anxiety. The following Table 4 is to understand the comparison between several ethnic groups.

Recommendations for reference values are summarized in (Table 5).

### 7.1. Kidney Volume

The kidney volume is determined from the length, anteroposterior diameter, and depth using an ellipsoid formula (volume = length × width × thickness × π/6) [50]. Determining the volume in addition to the kidney length can be helpful in distinguishing between acute and chronic renal failure and in cases of acute transplant failure [27]. In autosomal dominant polycystic kidney disease (ADPKD), total kidney volume is a validated prognostic marker for risk assessment [51]. However, the kidney rarely conforms to a true ellipsoid shape, and hence, it can lead to underestimation of true renal volume, leading to imaging inaccuracies and reduced reliability. In addition, ultrasound-based volumetry is limited due to high inter-operator variability and sub-optimal visualization due to known limitations such as bowel gas, ribs, absent perirenal fat, or challenging body habitus (obesity or central adiposity). Also, compared to MRI, the accuracy and reproducibility of the kidney volume achieved are somewhat lower. Hence, limits the utility in settings that require precise volumetric measurements.

### 7.2. Too Large or Too Small?

Findings are derived from the overall assessment of the kidney, the longitudinal diameter, the width of the parenchyma or cortex, the echogenicity of the parenchyma, and, if necessary, by calculating the kidney volume Table 6 provides an overview of when kidneys are too large or too small.

## 8. Echogenicity

The outer cortex is isoechoic or hypoechoic compared to the normal liver (right kidney) or normal spleen (left kidney). However, steatotic liver conditions can increase the liver brightness or echogenicity [52] and hence, become an unreliable reference standard for comparing the echogenicity relative to the kidney. In such a scenario, the spleen is preferred as a comparator organ. The medullary pyramids are hypoechoic or anechoic compared to cortex [9]. These pyramids terminate at the renal papilla, which protrudes into the minor calyces (Table 7). In adults, the fusion of these lobules renders them indistinct on imaging. It is important to mention that until age of six months, due to cellularity and water content, the cortex is hyperechoic than the liver and spleen [10].

Renal echogenicity is a key ultrasound marker for assessing renal health. It is important to note that the status of hydration, ultrasound machine settings, and occult liver pathologies (steatosis, cirrhosis-increased echogenicity) can affect the baseline echogenicity of the kidneys. An increase in cortical echogenicity and renal pathology is correlated. Moghazi et al. and Page et al. associated it with advanced glomerular or tubular damage and glomerulosclerosis, respectively [53]. Loss of corticomedullary differentiation (when medulla and cortex are indistinguishable) can indicate an infection or renal vein thrombosis in native kidneys and rejection in a transplanted kidney [42]. The list of renal pathologies and their parenchymal echogenicity findings is summarized in Table 8 [54].

### 8.1. Cortical Thickness

Cortical thickness is measured sonographically from the outer border of the medullary pyramids or arcuate arteries to the renal capsule. Normal values based on arteriographic data state that the range varies between 7.5–9.5 mm and 5.0–8.0 mm in healthy adults [55]. it is important to acknowledge potential inter-observer differences due to imaging planes and general body habitus that can lead to variation in the values [4]. Prior studies have shown that transverse axial cortical thickness may serve as a sensitive marker for pathological disease, including atherosclerotic renal artery disease. Although the measurements’ consistency can be operator- and center-dependent, emerging evidence supports its utility in clinical decision making [56]. Changes in cortical thickness in patients who underwent transcatheter aortic valve replacement (TAVR), indicate a potential role in predicting renal outcomes in these patients [57]. Nwafor et al. performed a comparative sonographic assessment of 150 hypertensive and equal number of normotensive patients. The study concluded that the mean cortical thickness was 1.0 ± 0.2 cm bilaterally, while normotensives exhibited higher values of 1.2 ± 0.2 cm on the right and 1.3 ± 0.2 cm on the left kidney. These findings highlight how cortical thickness is a non-invasive, reliable parameter for identifying renal damage in hypertensive patients, even in the absence of overt functional decline [58].

### 8.2. Parenchymal Thickness

It is described as the distance from the outer parenchymal contour of the kidney cortex to the tip of the medullary pyramid, measured at a right angle. Its approximate measurement is >15 to 16 mm in adults, though there is no definitive normal range. Parenchymal thickness decreases with an increase in age, with approximately a 10% decrease per decade [59]. Parenchymal thinning below 10 mm has been associated with severe renal damage, particularly with the history of chronic pathologies (chronic kidney disease, diabetic nephropathy, and hypertensive nephropathy) [60,61,62].

### 8.3. Cortico Medullary Ratio

This is defined as the ratio of cortical thickness to the length of the adjacent medullary pyramid. In healthy transplanted donors, the ratio was shown to be a mean of 0.97 with a range of 1.4 to 1.5 [63]. Arteriographic studies have reported a narrow normal range of 0.81–1.0 [64]. In a clinical setting, its utility is limited.

### 8.4. Congenital Variations and Anomalies

There are congenital changes that have no effect on kidney function and are primarily of differential diagnostic significance. Other congenital anomalies, such as polycystic kidney regeneration, lead to end-stage renal failure. Congenital changes and malformation affect kidney position, number, contour, parenchyma, malfusion, malrotation, the urinary tract, and congenital kidney cysts (Table 9).

## 9. Doppler Ultrasound (Vasculature Assessment)

Doppler ultrasound utilizes color Doppler, power Doppler, and spectral Doppler modalities to characterize blood flow dynamics. Color Doppler imaging provides real-time visualization of flow direction and velocity, while power Doppler enhances sensitivity for detecting low-velocity flow, independent of flow directionality. Spectral Doppler analysis offers a graphical representation of flow velocities over time, facilitating the calculation of key hemodynamic indices including peak systolic velocity (PSV), resistive index (RI), acceleration time (AT), and acceleration index (AI) as seen in Figure 7. A commonly encountered phenomenon, aliasing occurs when the measured flow velocity exceeds the Nyquist limit, manifesting reversal of color encoding. Standardization of Doppler techniques is essential for reproducibility and reduced variability in measurements. Recommended measures are: (a) Maintaining a Doppler insonation angle of ≤60° for accurate results; (b) Placing a small volume at the origin of the renal artery, the middle portion, and the hilum, or at the intrarenal segmental branches; (c) Also, accounting for respiratory motion, which poses significant challenges in acquiring clear images. To mitigate this, targeting acquisition towards the end of expiration or a brief breath hold is helpful [11,21,66].

*Renal artery waveform characteristics:* The normal renal vascular architecture comprises the main renal artery, which branches into segmental, interlobar, arcuate, and interlobular arteries. Normal renal arteries exhibit a characteristic low-resistance waveform, defined by a rapid systolic upstroke, sustained forward diastolic flow, and minimal spectral broadening. Within the main renal artery, normal PSV values range between 60 and 100 cm/s, characterized by laminar flow. The RI is calculated as (PSV − end diastolic velocity)/PSV, and typically ranges from 0.56 to 0.70, reflecting optimal renal parenchymal perfusion. RI values in accessory renal artery territories are expected to be no different from those in other segmental arterial territories. An exception to this statement would be a truly aberrant segmental artery that originates from the superior mesenteric artery or intrarenal spermatic artery [52]. AT is the interval from systolic onset to peak velocity, which is normally less than 70 milliseconds (ms). AI is PSV divided by AT, and generally exceeds 3.0 m/s^2^ [11,21,67].

*Renal vein waveform characteristics*: The renal vein demonstrates continuous, low-velocity, monophasic flow, indicative of normal venous outflow. Quantitative assessment of venous hemodynamics is performed using the venous impedance index (VII), with elevated impedance indicating parenchymal fibrosis or increased interstitial pressures [11,21,67].

*Renal vascular pathology:* Recognition of deviations from these normative parameters is critical for the diagnosis of renal vascular pathology. Key diagnostic indicators include elevated PSV (>200 cm/s), spectral broadening, presence of parvus-tardus waveforms, absent or reversed diastolic arterial flow, and absence of venous flow signals. The following table summarizes key Doppler findings in commonly encountered pathologies (Table 10).

### 9.1. Masses

Detection of renal masses on ultrasound (US) is frequently an incidental finding. These masses are broadly categorized as simple cystic, complex cystic, or solid lesions, each with distinct sonographic characteristics.

Simple cysts appear as a round or oval lesion filled with anechoic content. A simple cyst has well-defined margins and an imperceptible wall with posterior acoustic enhancement. They are benign and do not require any further evaluation [73]. The incidence of renal masses increases with age, and there is a 50% chance of having a simple cyst in either of the kidneys by the age of 50 [10]. Complex cystic masses are kidney lesions that exhibit features such as septation, nodularity, and wall irregularities. They necessitate further characterization. This can be achieved through contrast administration and evaluation through contrast-enhanced ultrasound (CEUS) or contrast-enhanced computed tomography (CECT). Based on the complexity of cysts on imaging, they are categorized using the Bosniak classification system, which guides risk stratification and management (kindly refer to Appendix A for further description) [74,75,76].

Solid renal masses appear in the US exam with internal echoes, without a well-defined smooth wall as seen in cysts. These lesions may be benign or malignant (Table 11). One of the most common malignant lesions is renal cell carcinoma (RCC), which accounts for 80–90% of all malignant renal tumors of the kidney [77]. Other malignancies include urothelial carcinoma, lymphoma, and metastasis. Benign solid tumors include oncocytoma and angiomyolipoma. It is worth noting that the US may miss small renal masses. Some benign and malignant masses share overlapping features and hence necessitate further evaluation to establish a definitive diagnosis [78].

### 9.2. Kidney Stones

Ultrasound (US) is choice of initial imaging modality, particularly in pediatric patients (≤14 years old), pregnant patients, and patients with recurrent nephrolithiasis [79,80]. It effectively identifies stones in the calyces, renal pelvis, and ureteric junction (especially with a full bladder) [81]. Approximately 30–50% of kidney stones are asymptomatic and are incidentally detected on US [82]. While smaller stones ≤ 3 mm may be missed on US, larger stones ≥ 5–7 mm can be reliably detected, especially if there is an accompanying hydronephrosis [10]. On grayscale US, stones appear as hyperechoic foci with a posterior non-echogenic shadow [80]. Additionally, in Doppler US, twinkling artifacts are often seen below the stone, which can further aid detection [82]. Larger stones, such as coral or staghorn calculi, fill the collecting system and are well visualized on US [7].

In an appropriate clinical setting, starting with an initial ultrasound procedure may reduce the need for a CT scan and limit radiation exposure. Also, a nondiagnostic ultrasound can always be followed by an alternative imaging modality, as recommended in the European Association of Urology (EAU) guidelines [83]. Ultrasound currently has some limitations, but efforts are underway to improve its effectiveness for kidney stone imaging.

## 10. Conclusions

Ultrasound is a frontline imaging modality in the evaluation of renal anatomy and pathology, offering real-time, non-invasive, and radiation-free assessment that is particularly valuable in both acute and chronic clinical settings. A detailed understanding of normal renal anatomy—including cortical thickness, parenchymal echogenicity, vascular anatomy, and kidney size relative to age, sex, and ethnicity—is critical for distinguishing physiological variations from pathological changes [84,85,86,87,88,89,90,91,92,93].

Standardized sonographic measurements are essential for early detection of conditions such as chronic kidney disease, renovascular hypertension, renal artery stenosis, and obstructive uropathy. Quantitative parameters like renal length, cortical and parenchymal thickness, and Doppler-derived resistive indices provide valuable diagnostic and prognostic information. Moreover, ultrasound serves as the initial diagnostic tool in evaluating renal masses and urolithiasis, reducing the need for more invasive or radiation-based imaging, particularly in pediatric and pregnant populations.

Clinicians should integrate ultrasound findings with laboratory data and clinical context to guide timely and accurate decision-making. Awareness of anatomical variants and population-specific reference values further enhances diagnostic confidence, particularly in diverse patient populations. As ultrasound technology continues to evolve, adherence to standardized scanning protocols and measurement techniques will remain central to maximizing its diagnostic yield in renal medicine.

## Figures and Tables

**Figure 1 diagnostics-15-03208-f001:**
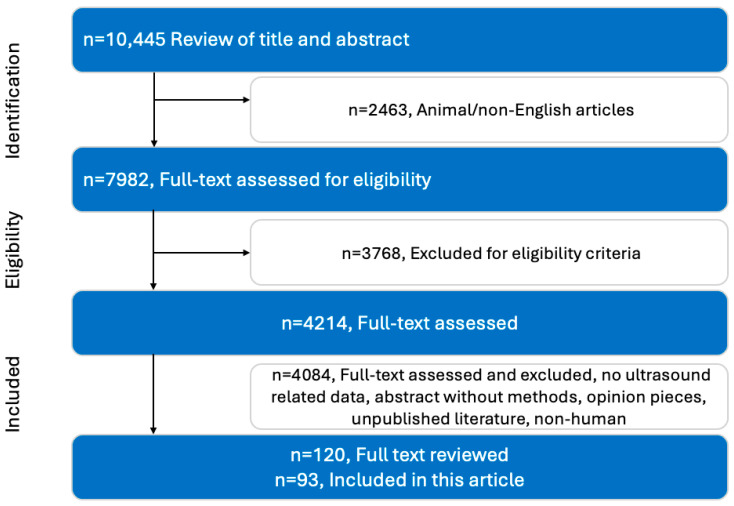
Flow chart depicting the search strategy employed and selection of studies finally included in this narrative review. In addition, publications identified through the reference lists of eligible articles, as well as relevant guidelines and papers necessary for understanding and explaining the topic, were also considered. Therefore, an additional 30 studies were included in analysis.

**Figure 2 diagnostics-15-03208-f002:**
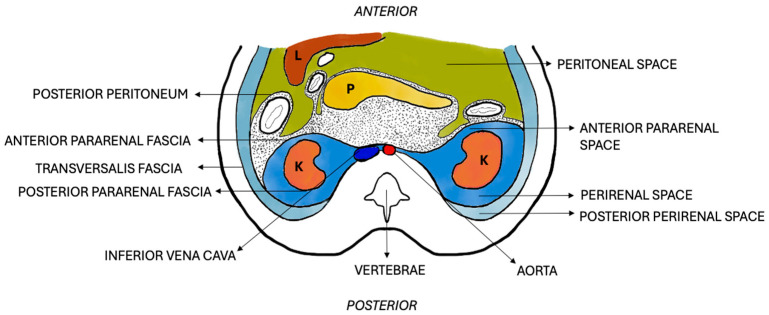
Schematic representation of transverse section through the abdomen showing the pararenal fasciae and spaces. (L: liver; K: kidney, and P: pancreas).

**Figure 3 diagnostics-15-03208-f003:**
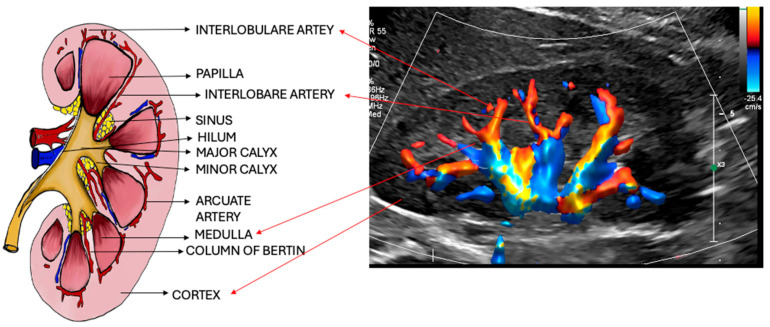
Schematic diagram of intrarenal anatomy in a midline coronal section.

**Figure 4 diagnostics-15-03208-f004:**
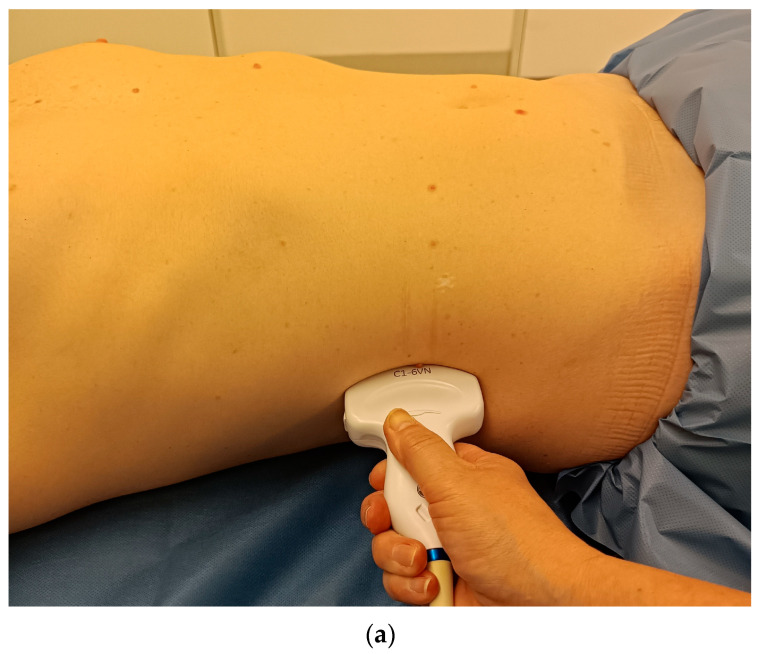
Right kidney in longitudinal section. Longitudinal transducer position in the area of the right anterior or midaxillary line (**a**). In the maximum extension, the pole distance is measured from the capsule of the upper kidney pole to the capsule of the lower kidney pole. Normal parenchymal echogenicity. The liver parenchyma is slightly more echogenic than the parenchyma of the healthy kidney. In the renal parenchyma, the medullary pyramids are heart-shaped and slightly less echogenic. The parenchymal thickness is measured vertically from the capsule to the tip of a medullary pyramid (**b**).

**Figure 5 diagnostics-15-03208-f005:**
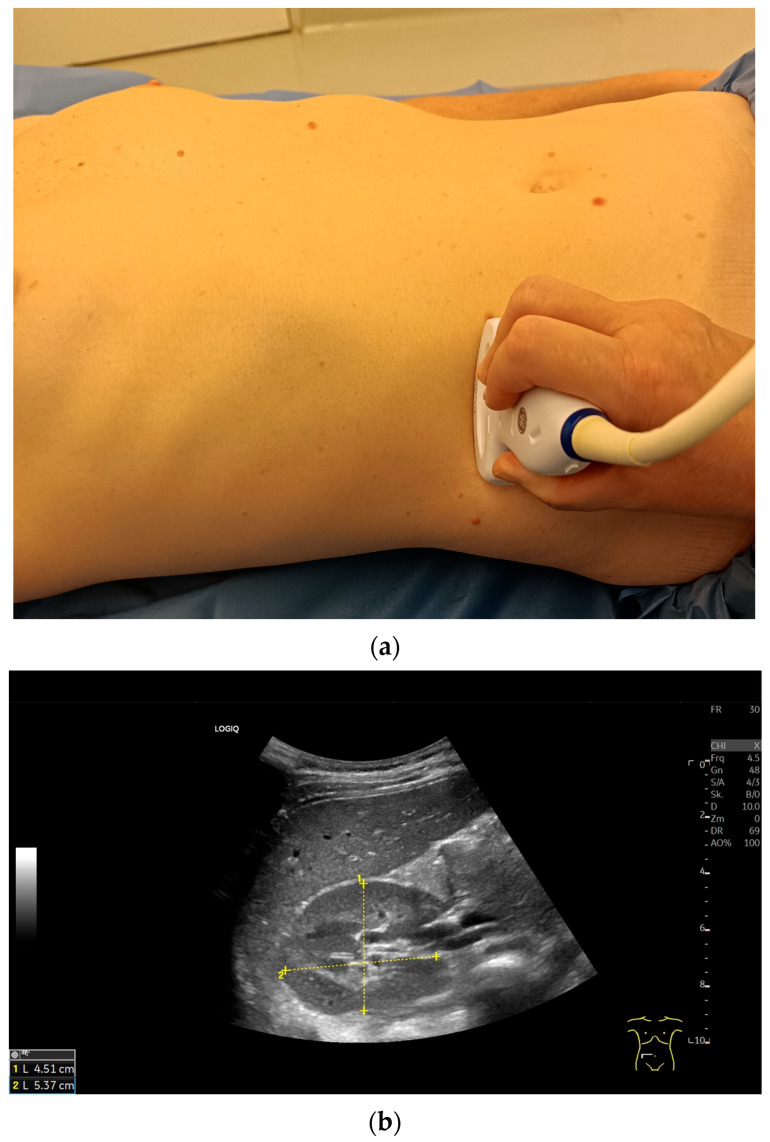
Right kidney in cross section. Transversal transducer position in the area of the right anterior or midaxillary line (**a**). The anteroposterior diameter is measured and transversely the depth from the hilus to the lateral outer contour (**b**).

**Figure 6 diagnostics-15-03208-f006:**
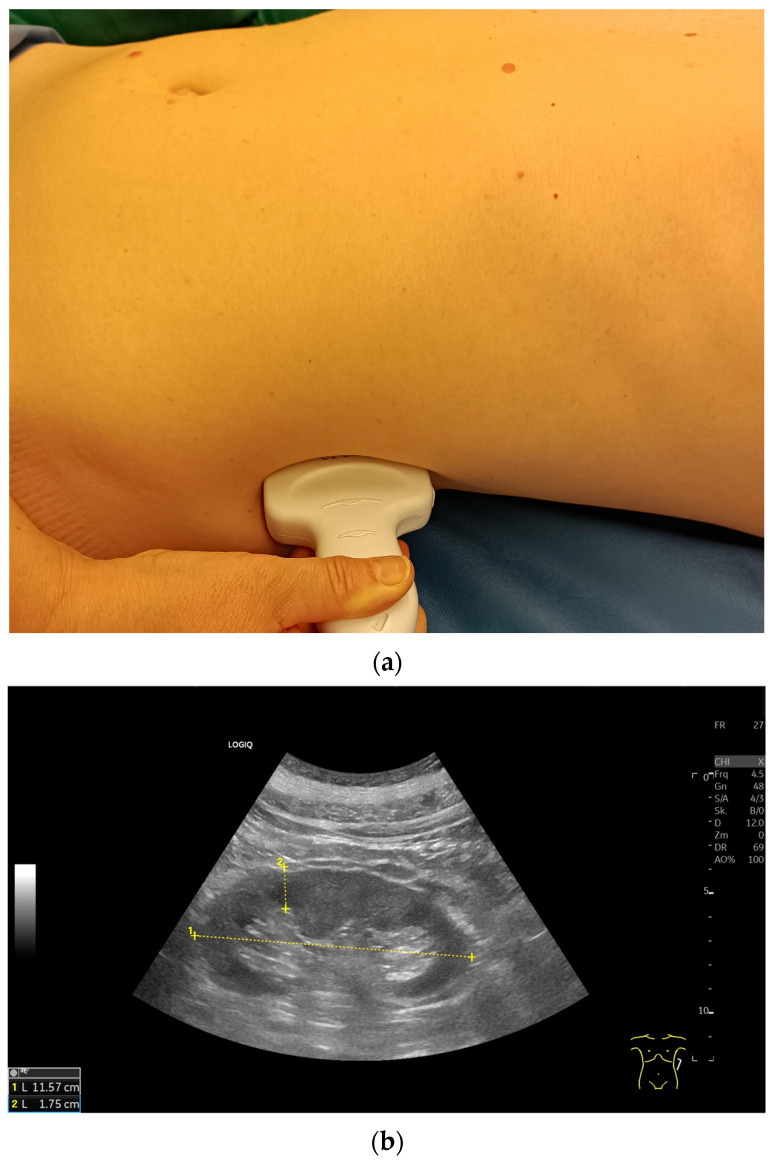
Measurements on the left kidney are performed as described on the right in longitudinal section (**a**,**b**) and transverse section (**c**,**d**).

**Figure 7 diagnostics-15-03208-f007:**
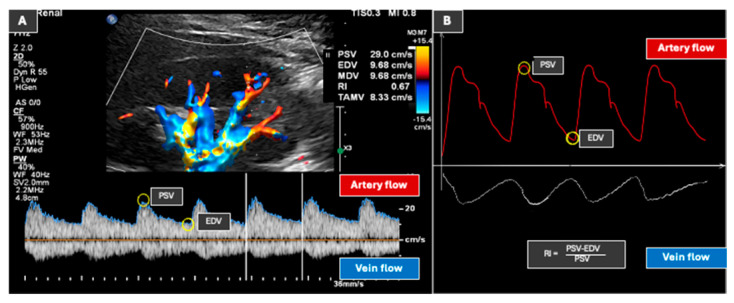
(**A**) depicts an image of normal renal arterial and venous spectral Doppler waveforms of the left kidney. (**B**) depicts an illustration of the spectral Doppler waveforms.

**Table 1 diagnostics-15-03208-t001:** Common patient positions used during renal ultrasound and their clinical purpose.

Patient Position	Clinical Purpose
Supine	Opens up rib spaces to facilitate better visualization
Lateral decubitus	To reduce bowel gas interface
Oblique	In obese patients
Prone	Occasionally used for posterior access
Standing	In cases of clinically relevant suspicion of kidney descent when standing.

**Table 2 diagnostics-15-03208-t002:** Recommended transducer types and frequencies for kidney ultrasound. The use of even higher-frequency transducers for newborns and very young children can be considered.

Patient Type	Transducer Type	Frequency Range	Notes
Adult	Curved array transducer	1–6 MHz	For deep imaging
Pediatric	Linear array transducer	Higher frequency (>9 MHz)	Better resolution for superficial kidneys

**Table 3 diagnostics-15-03208-t003:** What is measured and how.

Parameter	How to Measure
Length/maximum pole distance	Maximum length distance from the kidney contour at the upper pole to the contour at the lower pole. If the parenchymal margin at the upper and lower poles is of different thickness, the kidney is displayed tangentially.
Parenchymal thickness	From the outer contour to the tip of a medullary pyramid at a right angle.
Cortical thickness	From the outer contour to the base of the medullary pyramid or border between the cortex and medullary pyramid.

Please note: Throughout this review, we have consistently utilized the above definitions to describe cortical and parenchymal thickness.

**Table 4 diagnostics-15-03208-t004:** Comparison of renal parameters between different countries.

Author; Year (Reference)	Country	Sex	Kidney Length (cm)	Cortical Thickness (cm)
Wang et al., 1989 [41]	Malaysia (205)	M	R = 10.2	-
			L = 10.5	
		F	R = 11.2	-
			L = 11.5	
Emmanian et al., 1992 [29]	Denmark (665)	M	R = 11.0	-
			L = 11.0	
		F	R = 11.2	-
			L = 11.5	
		Both	R = 10.9	-
			L = 11.2	
Buchholz et al., 2000 [33]	Pakistan (194)	M	R = 10.6 ± 0.8	R = 1.6 ± 0.2
			L = 10.6 ± 0.8	L = 1.7 ± 0.2
		F	R = 10.3 ± 0.8	R = 1.5 ± 0.2
			L = 10.3 ± 0.8	L = 1.5 ± 0.2
Barton EN et al., 2000 [42]	Jamaica (39)	Both	R = 9.7 ± 0.7	-
			L = 10 ± 0.7	
Muthusami et al., 2001 [43]	India (140)	Both	R = 9.6 ± 0.97	* 1.4–2.7
			L = 9.71 ± 0.89	
Dominguez-Mija et al., 2001 [44]	Philippines (264)	M	R = 9.6	R = 0.42 L = 0.43
			L = 9.8	
		F	R = 9.5	R = 0.39 L = 0.39
			L = 9.3	
Hekmatnia A et al., 2004 [26]	Iran (230 M,170 F)	M	R = 11.0 ± 0.918	-
			L = 11.8 ± 1.04	
		F	R = 10.7 ± 0.637	-
			L = 10.9 ± 0.78	
Okoye IU et al., 2006 [45]	Nigeria (309)	Both	R = 10.33 ± 0.7	-
			L = 10.45 ± 0.63	
Oyuela-Carrasco et al., 2009 [46]	Mexico (153)	M	R = 10.57	N/A
			L = 10.72	
		F	R = 10.29	
			L = 10.46	
El-Reshaid et al., 2014 [47]	Kuwait (252)	M	R = 10.8 ± 0.9	R = 0.98 ± 0.2
			L = 10.9 ± 0.8	L = 1.02 ± 0.2
		F	R = 10.5 ± 1.1	R = 0.98 ± 0.8
			L = 11.2 ± 0.9	L = 1.02 ± 0.6
Su et al., 2019 [48]	China (3707)	M	R = 10.76 ± 0.66	R = 1.51 ± 0.31
			L = 10.87 ± 0.69	L = 1.39 ± 0.31
		F	R = 10.41 ± 0.67	R = 1.52 ± 0.29
			L = 10.59 ± 0.68	L = 1.45 ± 0.29
Khan SA et al., 2018 [39]	Pakistan (2212)	M	R = 10.30 ± 0.87	R=1.19 ± 0.12
			L = 10.38 ± 0.98	
		F	R = 10.18 ± 1.22	L = 1.26 ± 0.14
			L = 10.23 ± 0.92	
Tiryaki S et al., 2023 [40]	Turkey (1918)	M	R = 11.01 ± 0.72	^#^ R = 1.52 ± 0.16
			L = 11.38 ± 0.74	L = 1.56 ± 0.16
		F	R = 10.45 ± 0.65	R = 1.42 ± 0.14
			L = 10.75 ± 0.65	L = 1.45 ± 0.13
Ali S Aliyami et al., 2024 [49]	Saudi Arabia (55 M, 40 F)	M	R = 9.96 ± 1.21	-
			L = 10.4 ± 0.78	
		F	R = 9.53 ± 0.636	-
			L = 9.64 ± 1.14	

* Parenchymal thickness. ^#^ The article illustration shows the distance between the renal cortex and the renal sinus, depicting parenchymal thickness rather than cortical thickness (distance between the cortex and the base of the renal medullary pyramid).

**Table 5 diagnostics-15-03208-t005:** Reference values.

Parameter	Reference Values
Length/maximum pole distance	10–12 cm
Parenchymal thickness	>15 mm
Cortical thickness	>5 mm

**Table 6 diagnostics-15-03208-t006:** When are kidneys too large or too small?

Too Large	Too Small
Double kidney	*Unilateral:*
Compensatory hypertrophy in single kidney	congenital dysplasia
Pregnancy	renal artery stenosis
Early-stage diabetic nephropathy	*Bilateral:*
Acute nephritis	Chronic glomerulonephritis
Acute renal failure	Chronic pyelonephritis
Transplant rejection	Advanced diabetic nephropathy
Right heart failure with venous congestion and retrograde flow	Advanced arteriolonephrosclerosis
Renal vein thrombosis	Shrinking kidneys in end-stage renal failure
Urinary retention	
Crush injury	
Amyloidosis	

**Table 7 diagnostics-15-03208-t007:** Ultrasound echogenic characteristics of intrarenal structures.

Intrarenal Structure	Ultrasound Appearance
Cortex	Isoechoic or hypoechoic compared to non steatotic liver/spleen
Medullary Pyramid	Hypoechoic
Sinus Fat	Brightly echogenic
Renal Column of Bertin	Similarly to cortex; Continuous with cortex
Arcuate Vessels	Hyperechoic dots at -medullary medullary border
Collecting System	Not visualized unless distended with urine

**Table 8 diagnostics-15-03208-t008:** Common renal pathologies and their echogenicity findings.

Renal Echogenicity	
Hyperechoic	Hypoechoic
Chronic kidney disease	Cortical necrosis
Acute interstitial nephritis	Hemorrhagic infarcts
Amyloidosis	Lymphoma

**Table 9 diagnostics-15-03208-t009:** Congenital changes and malformations.

Congenital Changes in the Kidneys		
Nature of Changes	Description	Meaning
Changes in the Renal Surface		
Fetal lobulation “Renculation”	Uniform contour retractions over the columnae renales. Regular in neonates and very young children, rare in adults.	Can be confused (in adults) with scars.
Physiological spleen hump left	Protrusion at the outer contour of the parenchyma in the middle third. Parenchymal architecture with medullary pyramids and columnae renales and vascular branching in the parenchyma is preserved.	Misinterpretation of is echogenic tumors and parenchymal swelling (hematoma, melting-in inflammation).
Retraction at the parenchyma of the right kidney close to the liver	Transducer position and sonic angle-related retraction in the renal hilus.	Misinterpretation as a wedge-shaped scar.
**Parenchymal changes**		
Parenchyma cone (Bertin’s column)	Hypertrophied columnae renales, traversing vessels are preserved. Lateral to the parenchymal cones are the medullary pyramids. Parenchymal cones may be located centrally or asymmetrically.	Misinterpretation as echogenic tumors, especially if echogenicity is altered due to artifact.
Parenchymal bridges	The renal sinus is divided by one (or more) parenchymal bridges.	In itself, this is a typical finding. Part of a double kidney.
**Kidney malformations**		
** *Congenital changes in number and position* **		
Agenesia, Aplasia	Unapplied kidney.	Differential diagnosis to the condition after nephrectomy or to the dystopic kidney in other localization.
Hypoplasia	Small kidney with smooth contour and preserved architecture. Rule one-sided.	Differential diagnosis from renal artery stenosis with unilateral renal reduction, from unilateral shrunken kidney with altered, usually poorly demarcated kidney. Chronic kidney disease is usually associated with bilateral reduction in size.
Dystopia, Malposition (Ectopy)	The kidney is not in the usual position. Thoracic position in newborns, which impedes respiratory activity and must be quickly corrected surgically. Low kidney, low lumbar position, pelvic position.	If the kidney is not found, it is necessary to search in other positions. Common association with malrotation and fusion abnormalities.
Malrotation	The kidney is rotated in its axes. The kidney may be at a different angle and the hilus may be rotated. Association with other fusion anomalies and congenital kidney changes.	The kidney may not be recognizable and tumor-like images may be seen.
** *Fusion anomalies* **		
Double kidney	Large kidneys with parenchymal bridge and contour retraction above the parenchymal bridge with division of the renal sinus, including the renal pelvis. One kidney remains, not a true division into two kidneys.	Usually no diagnostic difficulty.
Horseshoe kidney	Large kidneys on both sides, usually double kidneys, parenchymal bridges. Often deeper located. The caudal pole is displaced medially and the inferior poles of both kidneys fuse over the aorta.	Solid mass over the aorta may be misinterpreted as a colon tumor.
Pelvic kidney	Pelvic location of a kidney or fused kidney.	Can be misinterpreted as a solid tumor. Distinguished from congenital pelvic position must be atypical mobility: drop kidney or ren mobilis.
Sigmoid kidney	One half or part of the kidney is rotated around the axis and fused with the other part. Parenchyma is adjacent to sinus and vice versa.	Tumor-like images are formed, triggering unnecessary diagnostics.
Cake kidney	In a pie kidney, both kidneys are fused into one and lie in front of the os saccrum. There is only one ureter. In the pie kidney, there is megacalicosis with dilated calyces.	The kidney of the cake can be confused with a (cystic) solid tumor.
**Urinary flow disorders**		
Ectopic ureteral orifice with obstruction	Ureteral dilatation	Urinary retention
Ectopic ureteral orifice with reflux syndrome	Visualization by retrograde radiological contrast or intravesical ultrasound contrast application with visualization of the contrast agent in the renal sinus.	Frequent urinary tract infections.
Crossing of the ureters	The ureters cross and open on the opposite side. Crossed ectopy with or without fusion anomaly with the ipsilateral kidney; solitary (unilateral) ectopy; or bilateral ectopy [65].	Cannot be visualized on ultrasound without visualization of the course of the ureters. Combination with other renal malformations/fusional relicts.
Ureteral outlet stenosis	Congenital constriction at the transition from the pyelon to the ureter. This can also be caused by transverse vessels.	Ureteral outlet stenosis can lead to hydronephrosis and impaired kidney function if it goes undetected.
Calyx diverticulum	Diverticula of the renal calices, appear like cysts.	Cannot be distinguished from cysts on ultrasound without excretory urography.
Megacalix	The collecting tube and the papillae are missing. Flared calyxes without congestion of the calyx necks and the renal pelvis.	Differentiation from urinary retention and renal cysts.
**Congenital cysts**		
***Primary cysts (malformations***)		
** *Polycystic kidney degeneration* **		
Adult familial cystic kidneys/adult form (autosomal dominant)	Enlarged kidneys with multiple cysts such that the original parenchyma and renal sinus cannot be delineated. Cysts may also occur in the liver and pancreas.	Terminal renal failure in adulthood. Familial clustering.
Juvenile cystic kidneys/from birth (autosomal recessive)	Cystic kidneys completely degenerated from birth, thousands of tiny cysts of a few millimeters.	Terminal renal failure in childhood

**Table 10 diagnostics-15-03208-t010:** Key Doppler findings of common renal vascular pathologies [11,21,66,68,69,70,71,72].

Renal Pathologies	Doppler Findings
Renal artery stenosis	Color saturation towards white, aliasing phenomenon, or a reversal in flow direction. Absence of flow signal (complete occlusion). PSV > 200 cm/s (compatible with ≥60% stenosis) *, increased renal/aortic PSV ratio (>3.5:1). “Parvus-tardus” waveform (blunted and delayed systolic upstroke), AI < 3 m/s^2^, AT > 70 ms.
Renal artery thrombosis	Parvus-tardus waveform or absent intra-renal Doppler signal.
Renal artery aneurysm	Grayscale US: anechoic mass and Doppler: continuity with the main renal artery (calcification and mural thrombus can mask flow signals).
Pseudoaneurysm	‘Yin-yang’ sign reflecting turbulent or swirling movement of blood flow.
Post-biopsy arterio-venous fistula (AVF	Aliasing phenomenon and a high-velocity mosaic pattern. Elevated systolic diastolic velocities with markedly reduced resistive index (0.30–0.40) in the feeding artery, and continuous high-velocity flow with diminished modulation in the draining vein.
Renal vein thrombosis or external compression	Partial obstruction: filling defect. Complete obstruction: complete absence of flow. Reversal of diastolic flow in the renal artery and an elevated arterial resistive index.

* Sensitivity and specificity ranging between 80~90% and 75~90%, respectively. These absolute velocity thresholds are subjected to variation (increases in: high-output states, anemia, fever, pregnancy, or tachycardia, and decreases in: low-output states, advanced heart failure, hypotension) and hence should be interpreted cautiously [35,66,69].

**Table 11 diagnostics-15-03208-t011:** The table summarizes sonographic features of some of the common solid renal masses [21].

Renal Mass			Imaging Features
Malignant solid masses	RCC		May have both solid and cystic components. Solid component: isoechoic, hypoechoic, or hyperechoic to the background parenchyma.
	Urothelial carcinoma		Typically, hypoechoic appearance. May appear slightly hyperechoic to the renal cortex. But more hypoechoic than the renal sinus fat.
Benign solid masses	Oncocytoma		Varying ultrasound appearance (isoechoic, hypoechoic, or hyperechoic). Color Doppler flow imaging shows “spoke-wheel” sign [62].
	Angiomyolipoma (AML)	AML Classic	Hyperechoic to renal parenchyma (lipid rich or classic AML).
		AML	Isoechoic to renal parenchyma (lipid poor AML).

## Data Availability

The original contributions presented in this study are included in the article/Appendix A. Further inquiries can be directed to the corresponding authors.

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
