# Peer review of "Sonographic Anatomy and Normal Measurements of the Human Kidneys: A Comprehensive Review"

_diagnostics, 2025, doi:10.3390/diagnostics15243208_

Round 1

Reviewer 1 Report

Comments and Suggestions for Authors

Dear Authors,

Firstly, thank you for submitting this comprehensive review on renal ultrasonography. The manuscript addresses a clinically important topic—normal renal morphology and measurements, echogenicity patterns, and Doppler parameters—and seeks to synthesize reference ranges and practical decision points for daily practice. The overall structure is logical, the figures and tables are useful, and the clinical orientation will be valuable for trainees and non-specialists.

Secondly, the strengths deserve emphasis. You gather normal ranges and anatomical variants across many decades and population subsets, and you frame the interpretation of cortical/parenchymal thickness, kidney length, and volume together with Doppler indices such as PSV, RI, AT, and AI. This dual focus on morphology and hemodynamics is appropriate for an educational review. Your statements about the role and limits of conventional B-mode/Doppler and the indications for escalation to CEUS are clinically sensible and align with contemporary practice.

Thirdly, the methods section requires greater transparency to achieve reproducibility and to align with COPE/PRISMA expectations for literature reviews. You currently describe a “systematic literature review” conducted between March 2024 and May 2025, while also summarizing evidence dating back to the 1950s. Please provide full Boolean search strings for each database, list all databases actually searched, and include a PRISMA-style flow diagram with explicit numbers (identified, screened, excluded with reasons, and included). If the approach is best characterized as a narrative review with systematic searching rather than a full systematic review, please adopt that terminology consistently across the abstract, methods, figures, and conclusions.

Fourthly, wherever you present “normal ranges,” please enrich the measurement context so readers can interpret values correctly. Specify the population characteristics (age distribution, sex, BMI, ethnicity), the equipment and technique (transducer frequency, patient position, exact measurement plane), and the uncertainty metrics (e.g., 95% confidence intervals or inter-observer variability). For cortical versus parenchymal thickness, define the terms consistently and explain how you measured them. For renal volume, explicitly state the geometric assumption (e.g., ellipsoid formula) and discuss typical sources of error in ultrasound volume estimation.

Fifthly, the Doppler section would benefit from a concise standardization panel to improve bedside reproducibility. Please summarize angle correction, sample volume placement (main renal artery versus segmental/arcuate branches), respiratory phase, and common pitfalls such as aliasing or high-output states. When you cite reference ranges and diagnostic thresholds (e.g., for suspected renal artery stenosis), add brief notes on sensitivity/specificity where available, and caution readers about clinical scenarios that shift absolute velocities (e.g., anemia, tachycardia).

Sixthly, your discussion of echogenicity and corticomedullary differentiation is clinically helpful, but a brief note on comparator organ caveats will prevent misinterpretation. In patients with steatosis or chronic liver disease, the liver may be unreliable as a reference for “increased echogenicity,” whereas the spleen can be preferable; stating this explicitly, supported by a representative image, would strengthen the section and enhance teaching value.

Seventhly, for indeterminate cystic or complex renal lesions, translating the narrative into a simple stepwise algorithm will increase clinical utility. A figure that begins with B-mode/Doppler findings, indicates when CEUS or CECT is recommended, and then links to classification/management pathways will help junior readers and non-radiologists apply your review at the point of care.

Eighthly, please harmonize terminology and units across text, tables, and figure captions. Use one term consistently for cortical/parenchymal thickness, ensure units for Doppler parameters are uniform (e.g., PSV in cm/s, AT in ms, AI with correct dimensions), and cross-check table notes so definitions do not drift between sections. Minor language edits for concision and active voice will also improve readability, especially in the Methods and normal-values sections.

Lastly, ensure that your ethics and data statements are appropriate for a review. If no unpublished patient data or identifying images are included, “Not applicable” is suitable for informed consent. If any third-party images or schematics are reproduced, please confirm permissions and cite the source explicitly. A brief data-availability note indicating that search strings and selection flow are provided in supplementary material will increase transparency.

In summary, the manuscript is close to becoming a practical reference for renal ultrasound. Strengthening methodological transparency, expanding the context and uncertainty around “normal” measurements, standardizing Doppler technique, and aligning terminology will materially improve rigor and educational value.

Sincerely,

Comments on the Quality of English Language

The manuscript is written in generally clear and professional English. However, certain sections—particularly in the Methods and Results—would benefit from minor editing for conciseness, consistent terminology (e.g., “cortical thickness” vs. “parenchymal thickness”), and harmonization of measurement units. Overall, the English is understandable and does not impede comprehension, but careful language polishing will improve flow and readability.

Reviewer 2 Report

Comments and Suggestions for Authors

This work is aiming at a comprehensive review of sonographic anatomy and normal measurements of the human kidneys. Nevertheless uptodate  tools provide online stiffness measurements of the kidney. This elastographic approach should also show up in your "systematic review" with the given keywords. Please at least discuss this option. Your introduction and the conclusions will be very helpful for the community especially for beginners. Standardization and quality assessment play a keyrole in careful clinical diagnostics. You should add some insights from the literature on quality control and management in ultrasound handling. Your figures are mostly well prepared but i recommend to replace the view on the "patients" in figures 4-6 by sketches that can give more spatial orientation. It is the development of higher frequency probes that allowed for more detailed imaging of kidney parenchyma. Advances in Doppler ultrasound provided insights into segmental arterial flow patterns including resistive indices as an indirect measure of microcirculatory impedance. Those elevated values correlate with progressive organ failure and fibrosis. Your focus is on anatomy but you should at least stress the importance of functional parameters given by ultrasound imaging on the diagnosis of anatomical findings.

Reviewer 3 Report

Comments and Suggestions for Authors

This paper by Yadav et al is a relatively comprehensive review about the technique of renal ultrasound and the normal anatomical findings.  It more briefly discusses some of the pathological anatomy that can be identified with ultrasound, but this is not the focus of the paper and appropriately there is not much detail for these pathological findings.  I found this paper to be quite informative with this focus on ultrasound technique, normal anatomy, and how the anatomy is measures.  It was also quite readable.  I have no suggestions for improvement.

Round 2

Reviewer 1 Report

Comments and Suggestions for Authors

The majority of revisions have addressed the points I requested. In particular, improvements in methodological transparency, terminology standardization, and Doppler technical details have significantly strengthened the article. The statistical context of normal values and the schematic algorithm for complex cystic lesions are still areas for improvement; however, the study is publishable in its current form.

Author Response

We kindly thank the Reviewer. Please see also the comments to the Editor.